# A New Approach for Sensitive Characterization of Semiconductor Laser Beams Using Metal-Semiconductor Thermocouples

**DOI:** 10.3390/s22239324

**Published:** 2022-11-30

**Authors:** Anna Katarzyna Piotrowska, Adam Łaszcz, Michał Zaborowski, Artur Broda, Dariusz Szmigiel

**Affiliations:** Łukasiewicz Research Network–Institute of Microelectronics and Photonics, Al. Lotników 32/46, 02-668 Warsaw, Poland

**Keywords:** nanothermocouple, FIB milling, micro- and nano-fabrication, Seebeck effect, laser beam characterization

## Abstract

This paper presents the results of beam investigations on semiconductor IR lasers using novel detectors based on thermocouples. The work covers the design, the fabrication of detectors, and the experimental validation of their sensitivity to IR radiation. The principle of operation of the manufactured detectors is based on the Seebeck effect (the temperature difference between hot and cold junctions induced voltage appearance). The devices were composed of several thermocouples arranged in a linear array. The nano- and microscale thermocouples (the hot junctions) were fabricated using a typical Si-compatible MEMS process enhanced with focused ion beam (FIB) milling. The performance of the hot junctions was tested, focusing on their sensitivity to IR radiation covering the near-infrared (NIR) radiation (λ = 976 nm). The output voltage was measured as a function of the detector position in the XY plane. The measurement results allowed for reconstructing the Gaussian-like intensity distribution of the incident light beam.

## 1. Introduction

In recent years, investigations on IR radiation have attracted much research attention due to its harmless effects on human health and ubiquity (emitted by objects near room temperature). Hence, it is applicable in various branches of human life such as, e.g., industry, military, medicine, environmental monitoring, telecommunication, biotechnology, etc. [1,2,3]. Nevertheless, some applications require IR radiation with specified parameters. The most valuable information on IR properties is provided by near-field analysis. Due to the limited and short distance covered by the near field, direct near-field measurements with commercially available detectors are challenging. Hence, the measurements are performed in the far field by applying additional light-focusing optomechanical components and commercially available detectors to measure the intensity [4].

Commercial detectors require additional cryogenic cooling, making them bulky, heavy, expensive, and inconvenient [5]. Unlike them, competitive thermal detectors do not require additional cooling and exhibit high sensitivity to radiation and short response time.

Nowadays, thermocouples are intensively used as components of OEM thermal–optical power detectors and thermal position detectors, which are commercially available [6]. Their common feature is a circle-shaped absorber interacting directly with incident light. Additionally, a top layer on the absorber, i.e., a blackened metal foil, is used to enhance absorption efficiency. Hence, the detectors are sensitive to a broad band of wavelengths (from UV to MIR). A laser operation mode defines the absorber performance [7]. For instance, detecting light emitted by long-pulse (0.1–10 ms) and CW lasers requires a surface absorber (ca. 0.1 µm–1 µm of thickness). On the other hand, the short-pulse (<1 µs) measurements need a volume absorber.

In commercially available power detectors, thermocouples surround the absorber. The output voltage is provided by the entire set of thermocouples (not only by a single one). Hence, the output voltage value depends on the number of thermocouples, which usually ranges from 20 to 120 and does not need an amplifier. However, additional parts/methods are required to prevent overheating the setup. The typical approach to preventing overheating involves a heat-conducting substrate (aluminum or copper).

Comparing commercially available thermoelectric detectors, the following issues should be considered:Responsivity;Specific detectivity;The thermoelectric figure of merit Z.

The most influential factors in the performance of a radiation thermoelectric detector are the Seebeck coefficient and the thermal time constant. High values are required for both coefficients. Thus, commercially available thermocouples consist of metal–metal junctions, such as bismuth and antimony. However, a metal–semiconductor junction exhibits several advantages over a metal–metal junction [8], i.e.,
The Seebeck coefficient of semiconductors is about one or two orders of magnitude higher than metals;The characteristics of semiconductor thermocouples are easily tunable by changing the concentration of doping;The micromachining techniques used in semiconductor processing allow the thermal capacity to be reduced effectively;Semiconductor-based thermocouples with high detectivity can be fabricated in a standard CMOS process.

A noteworthy application of a thermoelectric element (thermocouples) array to detect IR radiation was proposed by Wu et al. [9]. The reported system was a microscale spectrometer with a radiation thermoelectric detector and a diffraction grating fabricated on a standard chip. The thermocouples were arranged in a linear array, allowing an increased monitored area concerning the single thermocouple-based device.

In our work, we also applied the concept of a detector with a multipoint matrix composed of thermocouples arranged in the form of a ruler. However, to increase the sensitivity of the measurement, a metal (Ti/Pt) and semiconductor material (monocrystalline silicon substrate) were used as thermoelectric junctions. Detectors consisting of four and eight microjunctions were fabricated using the lithography approach, which was enhanced by focused ion beam (FIB) milling to obtain the nanoscale hot junctions. In this paper, we present the results of preliminary investigations on thermoelectric detectors that confirmed their high sensitivity to near-infrared (NIR) radiation. The experimental validation is preceded by a detailed description of the fabrication process of the thermoelectric detectors that will ultimately be used to characterize the laser infrared radiation.

## 2. Materials and Methods

### 2.1. Design and Fabrication

Thermoelectric detector chips were fabricated in CMOS-compatible technology using p-type 0.06 Ω cm silicon wafers acting as one of the thermocouple electrodes. The hot thermocouple junction and contacts to the Si substrate were designed in the center of a thin membrane that was etched beneath the thermocouple zone. This approach minimizes the heat capacity of the substrate, which improves the accuracy of thermometric measurements. The membranes were fabricated in the anisotropic etching of silicon in a hot KOH solution (Figure 1a). The etching depth was controlled by the optical inspection of test square cavities symmetrically located at the edges of the silicon wafer. Metal connection paths were made of aluminum on the oxidized silicon surface. The Al layer was sputter-deposited, etched by photolithography, and then sintered at 450 °C in a nitrogen gas shield. The hot thermocouple electrode was fabricated in a platinum layer with a thickness of 100 nm, which was deposited on a thin Ti layer to improve its adhesion to the substrate. Both layers were sputtered at 80 °C within one vacuum process. In the subsequent photolithography step, the Pt/Ti layers were etched in a newly developed plasma process in an ICP-RIE (inductively coupled plasma reactive ion etching) type reactor. The etching of platinum was carried out in Ar/Cl_2_ plasma at a temperature of 20 °C and pressure of 2 mTorr. The photoresist mask was removed in a microwave plasma reactor using O_2_ chemistry (Figure 1d).
Figure 1Cross sections at subsequent stages of fabrication of the thermoelectric detector before FIB treatment: (**a**) wafer oxidation, Si_3_N_4_ deposition, and silicon etching in KOH; (**b**) silicon nitride mask removal and Si membrane oxidation; (**c**) etching of contacts to the silicon; (**d**) sputter deposition of Al (cold contact to Si) and Pt metallization as a hot thermocouple. The sample structure of the detector is shown in the SEM image in Figure 2.
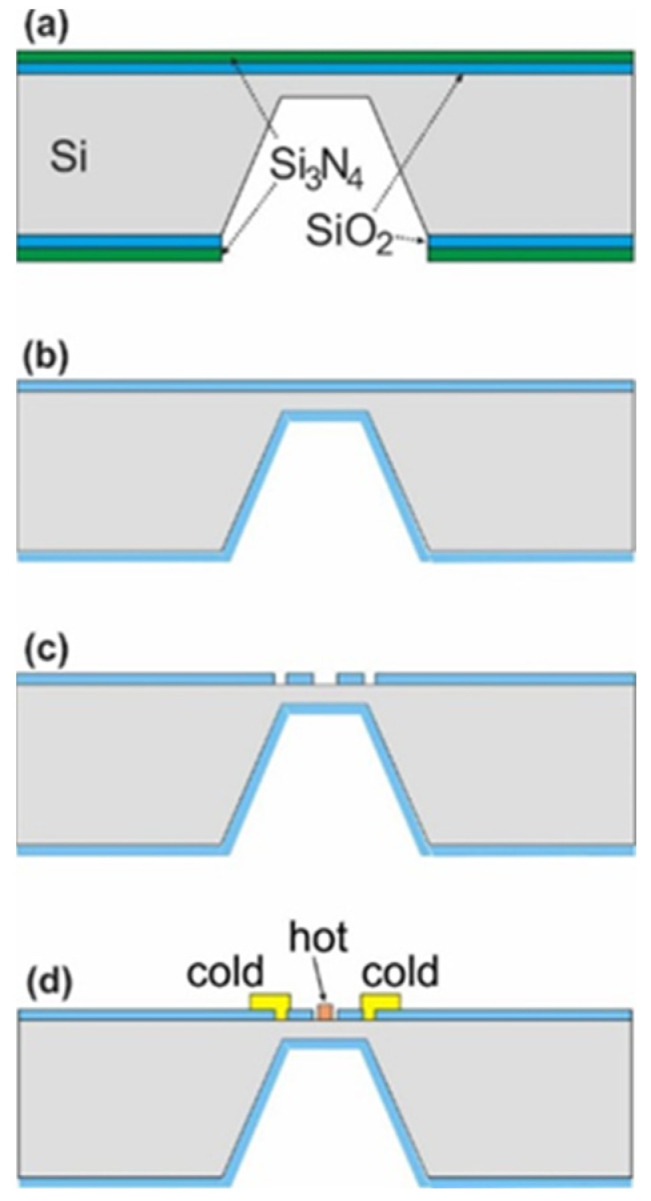

Figure 2SEM image of the thermometric structure (before FIB processing) showing the layout of cold (reference) and hot thermocouples.
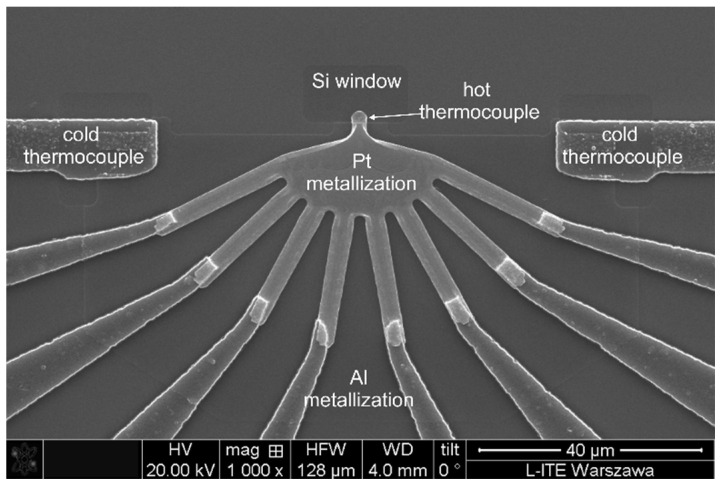



The 4-microthermocouples shown in Figure 3a were fabricated using photolithography techniques only, while the thermocouples presented in Figure 3b were prepared with FIB milling using the Helios 600 NanoLab DualBeam system. The FIB system is an excellent micro- and nanotechnology tool enabling easy prototyping [10,11] or post-processing customization of individual devices [12]. Thus, the micro- and nanostructures can be fabricated more precisely and much faster than when standard photolithography or imprint methods are applied.

The first step of the FIB process was depositing the SiOx layer (about 50 nm thick) from a gas precursor in a hot electrode area (Figure 2). This operation protected the sensitive surface of thermocouple strips from surface damage during ion milling. The oxide layer also served as an anti-reflecting coating in the finished device. The next step was an ion milling of the hot thermocouple area to form separated strips, according to the established milling pattern, designed for 8 strips with the desired width (about 400 nm) of the strips and the gaps between them. The last step was an ion milling of the remaining area of Pt metallization to fabricate separate electrical connections for each thermocouple. This way, the 8-thermocouple device was manufactured, as shown in Figure 3b. All FIB processes leading to the fabrication of such thermocouples were performed using ion-beam energy of 30 kV and an ion-beam current ranging from nanoamperes to picoamperes.

Silicon wafers with thermoelectric detectors with different sizes of thermocouples were cut into 6 mm × 6 mm dices and assembled on PCB substrates matching the ZIF connectors with a 0.5 mm standard. Appropriate dosing of an epoxy glue during assembling allows for equalization of pressures between the upper and lower side of the silicon membrane containing the complete device.

### 2.2. Experimental Validation

Two variants of multiprobe detectors were tested. One consisted of four microscale hot junctions (4 µm in width), and the other consisted of eight nanoscale hot junctions (400 nm in width).

It was assumed that only a part of the radiation energy causes the heating of the single hot junction, and it is strongly related to the position of the thermocouple in the illuminated area. Due to the Gaussian distribution of the incident light intensity, a different amount of energy interacts directly with the single thermocouple. Thus, each thermocouple measures a slightly different temperature within the area illuminated with the spot beam. The output voltage provided by the single thermocouple depends on the thermocouple’s position in the light spot. Hence, the Gaussian distribution of the illuminating light intensity may be reconstructed by measuring the related output voltage. Two approaches were undertaken to determine the voltage values. The single thermocouple was displaced in the illuminated area in the first one. Its position was defined by the displacement step from the initial position, i.e., ΔX = ΔY = 1 µm along the X axis and Y axis, respectively, for the microscale hot junctions and ΔX = ΔY = 0.625 µm for the nanoscale hot junction.

Figure 4a shows the light source of the experimental setup. This part consists of a fiber-coupled semiconductor diode laser emitting light with the wavelength λ_0_ = 976 nm. The light was guided by a single-mode fiber (1), which maintains the original profile of the light beam during its propagation. The fiber was connected to the fiber collimator (2) that formed the plane wave to keep the same light intensity over the light propagation. Two beam splitters were used to redirect the light toward the detector. The first one (3) redirected the white light to illuminate the investigated area, while the other (4) redirected the IR light beam. The IR beam was focused by the microscope objective with 40× magnification (5) and illuminated the detector (6). The power of the incident light beam measured by the power meter (digital handheld optical power and energy meter console, PM100D, THORLABS) immediately after leaving the microscope objective was equal to P_0_ = 300 µW. The spot diameter of the illuminated area was equal to D_0_ = 1 µm and covered the hot junctions. The cold junctions were not illuminated to prevent their heating. The CMOS camera (8) controlled the alignment of the light spot and the junctions. The detector was fixed to the stage (7), moving it in the XY plane.

In the first approach, the spatial resolution of the measurements referred to the displacement step along the *x*-axis and the *y*-axis (ΔX = ΔY = 1 µm for the microthermocouples, and ΔX = ΔY = 0.625 µm for the nanoscale thermocouples distribution).

## 3. Results and Discussion

The results of the measurements are shown as a function of the displacement of the thermoelectric detector. The displacement was measured with respect to the initial position of the detector.

The measurements reconstruct the Gaussian profile of the incident light beam (Figure 5a). It can be noticed that the highest output voltage is equal to ca. 11 mV, while the voltage measured with the nanoscale thermocouple is equal to ca. 5 mV. The small displacement step of the nanoscale thermocouples gives a higher spatial resolution of the measurements. Thus, a more detailed topology is shown in the picture (Figure 5d) representing measures taken with the nanoscale detector than with the microscale one (Figure 5b). The layout of the thermocouples is clearly recognizable in Figure 5d (the 2D graph) and reflects the original layout correctly.

The measurements allowed the successful reconstruction of the Gaussian shape of the incident light beam. Assuming thermal balance between thermocouples before the illumination, different output voltage values measured with each thermocouple refer to a different amount of energy provided by the incident light beam. Moreover, the plotted values can be involved in determining the temperature distribution on the surface illuminated by the incident light, as was done in the case of the multiprobe detector (Figure 6a,b).

Figure 6a,b depict the temperature difference between hot and cold junctions after illumination. The presented values were calculated based on the measured output voltage and the previously found value of the thermoelectric coefficient assessed at 0.54 mV/K for the thermocouple pSi-Ti/Pt. Furthermore, the 2D graph pictures the temperature distribution on the surface illuminated directly by the incident light in the area. The maximal value of the temperature increase induced by the incident light was equal to 10.9 K and corresponded to a measured output voltage of 5.88 mV.

The obtained results should be considered in qualitative, not quantitative, terms. The plotted values relate to the Gaussian distribution of incident light intensity, confirming the high sensitivity of the detectors to NIR radiation. However, the obtained results are not comparable with the ones provided by commercially available detectors, whose responsivity is the ratio of the output voltage (the sum of the voltages provided by every single thermocouple) to the average power of an incident beam. In our case, the output voltage is measured from every single thermocouple, and the incident power is a part of total power, illuminating the hot junctions directly. Furthermore, we have reported only the Seebeck coefficient, assuming that the thermal time constant is significant only for radiation emitted by pulse lasers, due to its influence on the radiation parameters such as repetition time, duty cycle, and pulse width.

Below, we compare market-available thermocouple sensors to the fabricated and studied devices (Table 1).

The authors have proposed junctions made of monocrystalline silicon and Ti/Pt with a much higher coefficient than junctions made of two dissimilar metals, which increased the sensitivity of the measurement. The developed thermoelectric detectors involved several independent thermocouples. It means that each thermocouple works as an independent sensor. Due to the small size, the thermocouples provided relatively small output voltage values, and additional signal amplification is needed. It should also be noted that the multiprobe sensitivity was successfully examined by the authors for continuous radiation and can also be used for pulsed radiation. Unlike them, the commercially available detectors are mainly dedicated to CW and quasi-CW radiation. Moreover, typical hot junctions are covered with an additional layer, increasing the absorption of incident radiation, which is unnecessary in the solution proposed by the authors. The most important advantage of proposed detectors is that they can be sensitive to the entire range of IR radiation, from near to far infrared. We also proved that the Pt/Si nanothermocouple detector made with the use of FIB records a thermoelectric signal even for visible light with a wavelength of 514 nm (green light) [11]. Resuming the investigations confirmed a significant advantage of the proposed detectors over the commercially available detectors.

## 4. Conclusions

We proposed a new approach to laser beam investigation on semiconductor IR-lasers using novel types of detectors based on thermocouples. The working principle of the detectors made of thermocouples arranged in a linear array relies on the Seebeck effect. The thermocouples (the hot junctions) were fabricated in micro- and nanoscale using a silicon-compatible MEMS process enhanced with the FIB milling to improve the resolution of the devices. The output voltages were measured as a function of the detector position in the XY plane. The characterization of the laser beam with the proposed detector was successful, allowing the reconstruction of the Gaussian-like intensity distribution of the incident light beam. Although the results should be considered in a qualitative manner, they enable temperature distribution at the surface of the detectors, corresponding to the power of the incident beam. The obtained results provide strong motivation to continue further research and development projects on the thermoelectric radiation detectors of optical radiation useful for the characterization of all types of laser beams in the near and far field.

## Figures and Tables

**Figure 3 sensors-22-09324-f003:**
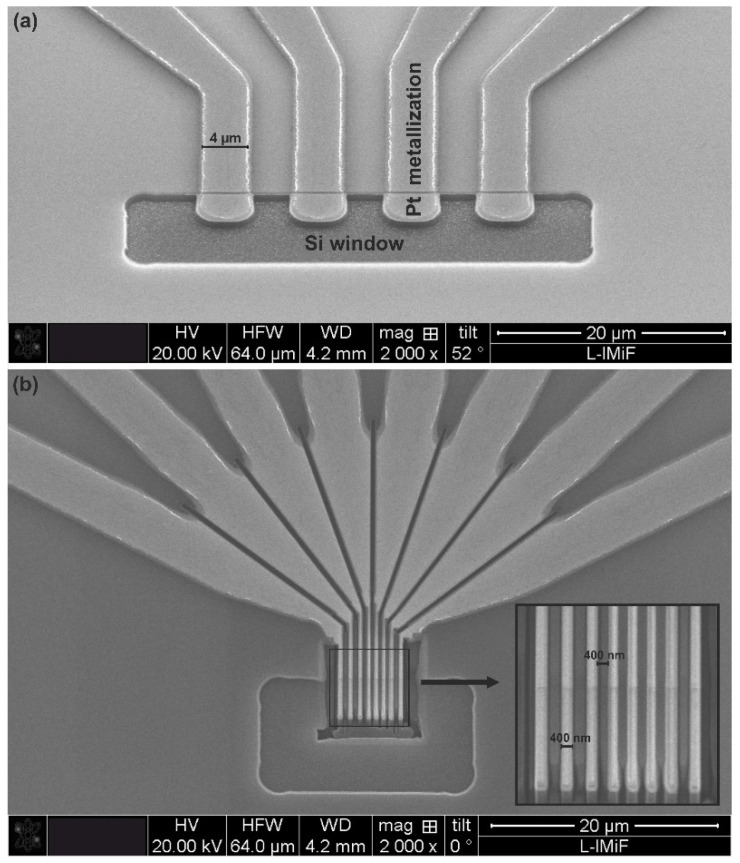
SEM image of the thermometric structures: (**a**) 4-microthermocouples (each 4 µm wide) prepared without using FIB; and (**b**) thermocouples prepared with FIB. The separate and narrow (400 nm width) 8-thermocouples (in the form of separated strips) were fabricated by milling using gallium ion. The air gap between thermocouples was about 400 nm. Each of the fabricated thermocouples has a separate electrical connection.

**Figure 4 sensors-22-09324-f004:**
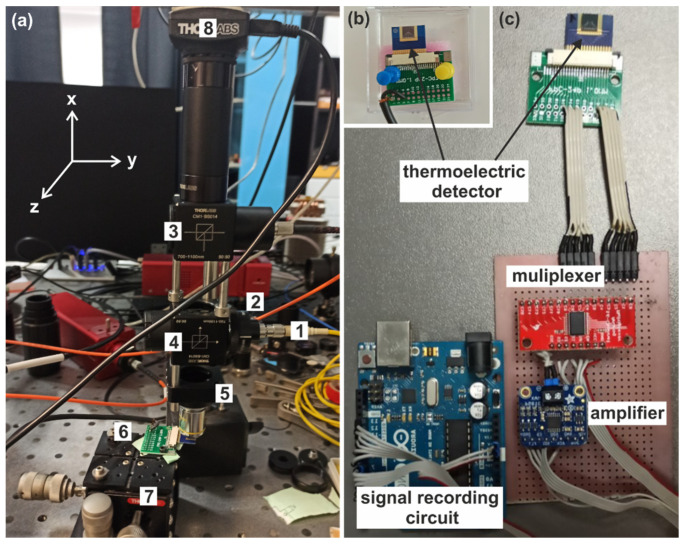
(**a**) The optical part of the experimental setup; (**b**) the configuration of the setup components to measure voltage output at a single thermocouple; and (**c**) the configuration of the setup components to measure voltage output at each available thermocouple simultaneously.

**Figure 5 sensors-22-09324-f005:**
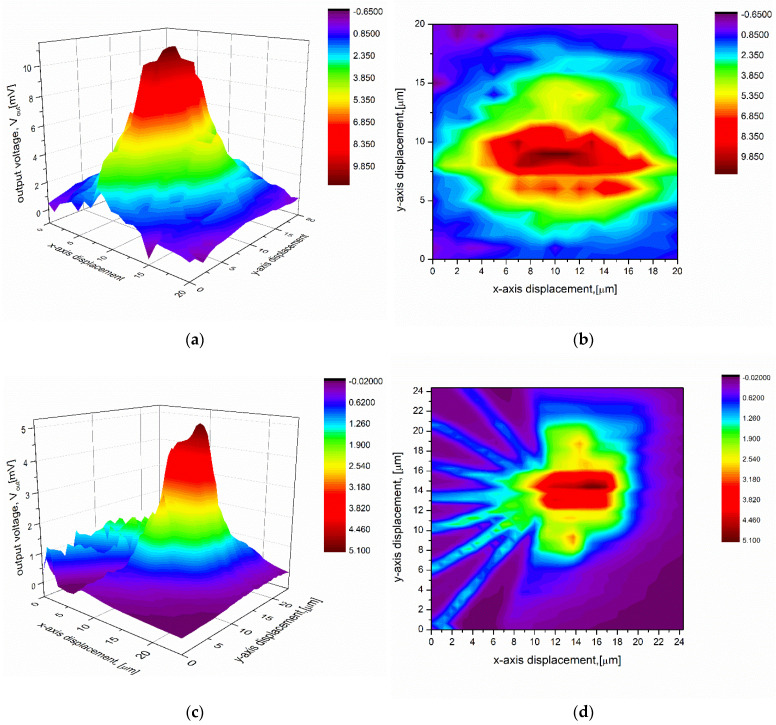
Measured thermoelectric voltages confirming the Gaussian distribution of the incident light intensity obtained using a single thermocouple with (**a**) the microscale hot junctions (shown in Figure 3a), and (**c**) the nanoscale hot junctions (shown in Figure 3b), displaced in the plane of the light spot; (**b**–**d**) top view of 3D graphs (**a**,**c**).

**Figure 6 sensors-22-09324-f006:**
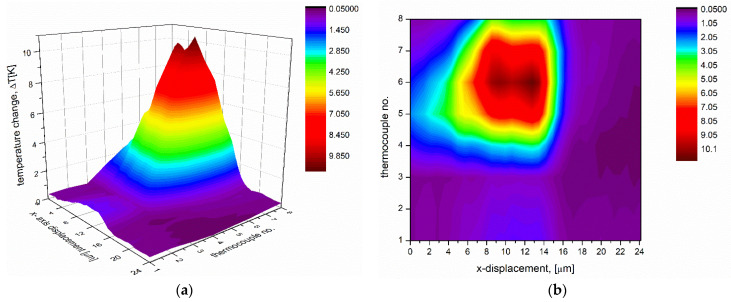
Two graphs plotted by the calculated values of the temperature increase based on the measurements performed by each thermocouple 1 to 8 (shown in Figure 3b) at the same time; (**a**) the 3D graph relates to the Gaussian profile of the incident beam; (**b**) the 2D graph depicts temperature distribution in the illuminated area.

**Table 1 sensors-22-09324-t001:** The comparison of market-available sensors and tested thermocouples.

Commercially Available Sensors [6,7]	The Developed Thermocouples
a junction of two dissimilar metals (usually bismuth and antimony)	the pSi-Ti/Pt junction
the output voltage is provided by a series of thermocouples (20–120 thermocouples) in a radial, tight arrangement	every single thermocouple provides the output voltage
usually dedicated to detecting CW and quasi-CW radiation	the detector was tested for detecting CW radiation but can be used also for pulsed radiation
additional cooling elements needed	no additional cooling needed
blackened hot junctions (surface layer to increase absorption)	no additional layer to enhance absorption is needed

## Data Availability

Not applicable.

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
