# Peer review of "A New Approach for Sensitive Characterization of Semiconductor Laser Beams Using Metal-Semiconductor Thermocouples"

_sensors, 2022, doi:10.3390/s22239324_

Round 1

Reviewer 1 Report

The following questions should be clarified before publication:

1.       In page 1 line 43, the grammar of the sentence “Hence, increasing interest in IR focal plane arrays 43 (FPAs) and their rapid development” should be corrected.

2.       In page 3 line 98, the brackets in the sentence “The na-97 noscale hot junctions enhance the spatial resolution of the measurements (the smaller the 98 hot junctions, the higher the spatial resolution of the measurements” were used incorrectly.

3.       How is the repeatability of the data as shown in figure 5?

4.       How significance is the output voltage (i.e., 11.15 mV) of the device compared to commercially available sensors? 

Author Response

Dear Reviewer,

Thank You for Your comments. Please, find our answers in the attached file.

Yours faithfully

Anna Piotrowska, PhD

Reviewer 2 Report

In this paperthe new approach for sensitive characterization of semiconductor laser beams using metal-semiconductor thermocouples was presented. The method is relatively novel, however, this article still needs to be greatly modified to meet the published requirements.

1.       The introduction part is a bit confusing, to it should be better organized to describe the significance of this work, the current research progress, the problems faced and the solutions proposed by the author.

2.       The results given by the author are mostly qualitative, and some quantitative parameters should be given, for example, what are the Seebeck coefficient and the thermal time constant of the sample?

3.       The overall structure of the paper is chaotic, and the structure of the whole article should be reorganized.

4.       The size of thin membrane in Figure 1 should be given.

5. In line 68,98 and 192, the punctuation brackets are not completely written.

Author Response

(The authors gave the same response as above.)

Round 2

Reviewer 2 Report

The author has made a good revision to the article, and the problems in the last draft have been solved. I agree that the article will be published in the current state.

Author Response

Thanks for your comments